materials science/nanotechnology

black phosphorus nanosheets, tribological properties, oil-based lubricant additive, titanium alloy

**Author for correspondence:**
Wei Wang
e-mail: gackmol@163.com

This article has been edited by the Royal Society of Chemistry, including the commissioning, peer review process and editorial aspects up to the point of acceptance.

This article has been edited by the Royal Society of Chemistry, including the commissioning, peer review process and editorial aspects up to the point of acceptance.

# Tribological properties of black phosphorus nanosheets as oil-based lubricant additives for titanium alloy-steel contacts

Qingjuan Wang[1], Tingli Hou[1], Wei Wang[1],
Guoliang Zhang[2,3], Yuan Gao[1] and Kuaishe Wang[1]

[1]School of Metallurgy Engineering, Xi'an University of Architecture and Technology, Xi'an 710055, People's Republic of China
[2]School of Mechanical Engineering, Tianjin University of Technology and Education, Tianjin 300222, People's Republic of China
[3]Guangdong Provincial Key Lab of Nano-Micro Materials Research, School of Chemical Biology and Biotechnology, Peking University Shenzhen Graduate School, Shenzhen 518055, People's Republic of China

WW, 0000-0002-6400-3599

The black phosphorus (BP) powders were prepared by high-energy ball milling with red phosphorus as the raw material, and then the BP nanosheets were obtained by liquid-phase exfoliation. The tribological properties of the BP nanosheets as oil-based lubricant additives were investigated by the ball-on-disc tribometer. Results show that compared with the base oil of liquid paraffin (LP), the coefficient of friction and wear rate of the BP nanosheets as the additives in liquid paraffin (BP-LP) are lower for the same loads. BP-LP lubricants could significantly improve the load-bearing capacity of the base oil for titanium alloy-steel contacts and show excellent friction-reducing and anti-wear properties. The surface morphologies and elemental compositions of the friction pairs were further analysed using an optional microscope, scanning electron microscope and X-ray photoelectron spectroscopy. The lubrication mechanism of BP-LP can be attributed to the synergistic effects between lamellar adsorption and interlayer shear of BP nanosheets.

## 1. Introduction

Titanium alloys are widely used in the aerospace, aviation, medical devices, military and automobile industries owing to low density, high strength at room and high temperatures, better

room temperature ductility, and excellent corrosion resistance [1,2]. However, titanium alloys are recognized as difficult-to-cut materials owing to their low thermal conductivity, high-temperature chemical activity and low modulus of elasticity [3,4]. When titanium alloys are sliding against most metals or ceramics, the surface of titanium alloys could easily produce severe adhesive wear [5,6]. Moreover, during the cutting process, a series of problems such as the adhesion wear between the workpiece and the tool, poor surface quality of the workpiece after cutting, and low machining precision limit the widespread application of titanium alloys in precision components [7,8]. Lubrication is the main method of solving the processing of titanium alloys. Owing to special tribological properties of titanium alloy, the traditional lubricants are not used in the processing of titanium alloys [8,9]. Therefore, the development of new lubricating material and methods with excellent tribological properties are the key to achieving the efficient lubrication in machine cutting of titanium alloy.

In the metal cutting process, lubricating additives with anti-wear and extreme pressure properties have played a very important role for improving the cutting ability. In recent years, two-dimensional (2D) materials such as graphene [10], molybdenum disulfide [11] and hexagonal boron nitride [12] have been widely studied in terms of lubricant additives with antiwear and extreme pressure properties owing to their excellent lubrication properties. In recent years, black phosphorus (BP) as a new 2D material has been widely investigated by domestic and foreign researchers, because of its special 2D layer structure and thermodynamic stability [13]. In addition, owing to its special microstructure and physico-chemical properties, BP has great potential for the application of friction and lubrication [14].

At present, a few studies have also shown the potential of BP nanomaterials in the field of tribology [15–19]. For instance, the micro-friction properties of BP nanosheets were evaluated by atomic force microscopy (AFM), the results showed that nanoscale friction of few-layer BP flakes were dependent on the layer number and friction presented anisotropy [15]. Based on the results of molecular dynamics simulation, micro-friction properties of BP nanosheets are highly anisotropic [16]. In addition, the environmental degradation of BP is also significantly beneficial to its lubricating behaviour [17]. In the field of macro-tribology, the macro-tribology behaviours of BP nanosheets as lubrication additives were also attractive to researchers. In our previous work, BP nanosheets modified by NaOH (BP-OH) as water-based lubrication additive could significantly reduce friction and achieve superlubricity [18]. BP nanosheets as the hexadecane-based lubrication additives also exhibited outstanding lubrication properties in the steel-steel contact. Compared with graphene oxide (GO) and $MoS_2$ nanosheets as hexadecane-based lubrication additives, BP nanosheets present similar tribological properties at lower loads and excellent extreme pressure and load carrying capacity at higher loads [14]. Yufu Xu et al. also researched the tribological properties of BP nanosheets in poly alpha olefin, the results showed that BP nanosheets have exhibited outstanding lubrication performance [19]. Lubricant additives with anti-wear and extreme pressure properties have an important role in the application of material processing, however, so far, BP as an oil-based additive has not been studied in the processing of titanium alloys.

In this paper, BP nanosheets were successfully fabricated through high-energy ball milling (HEBM) and liquid phase exfoliation methods. The tribological properties of BP-liquid paraffin (LP) lubricants were evaluated for titanium alloy-steel contacts. Moreover, the lubrication mechanisms were analysed based on the experimental results and discussion.

# 2. Experimental details

The experimental materials used in this investigation are red phosphorus (RP) (99.999%, Aladdin Reagent Factory), N-methyl pyrrolidone (NMP; greater than or equal to 99.5%), absolute ethanol ($C_2H_5OH$, AR reagent, 99.7%) and petroleum ether ($C_5H_{12}$, AR reagent). The LP was used as the base oil.

BP powders were prepared by the HEBM method. Firstly, RP and the stainless-steel balls ($\Phi$20 mm, $\Phi$10 mm) were added into the ball mill jar with a milling speed of 1200 r.p.m. and the ball-to-material ratio of 50 : 1 up to 2 h. After HEBM, the final powders were collected in a vacuum glove box filled with argon. Then the initial BP powders were dispersed in the NMP with ice bath for ultrasound treatment. The mixed dispersions were centrifuged at 3000 r.p.m. for 20 min to collect the supernatants. The supernatants were further centrifuged at 11 000 r.p.m. for 30 min to gather the precipitations. The BP nanosheets were obtained by repeated washing and drying. Finally, the BP nanosheets were added to LP, the BP-LP lubricants were prepared by ultrasound treatment for 2 h, as shown in figure 1.

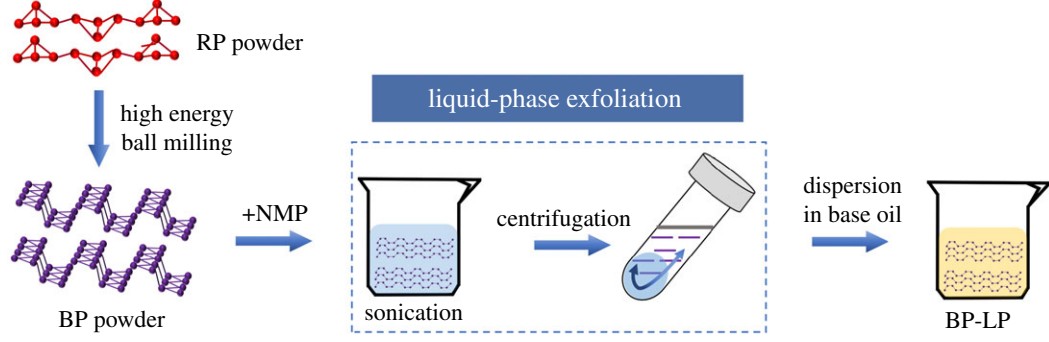

**Figure 1.** Schematic illustration of preparation of BP-LP lubricants.

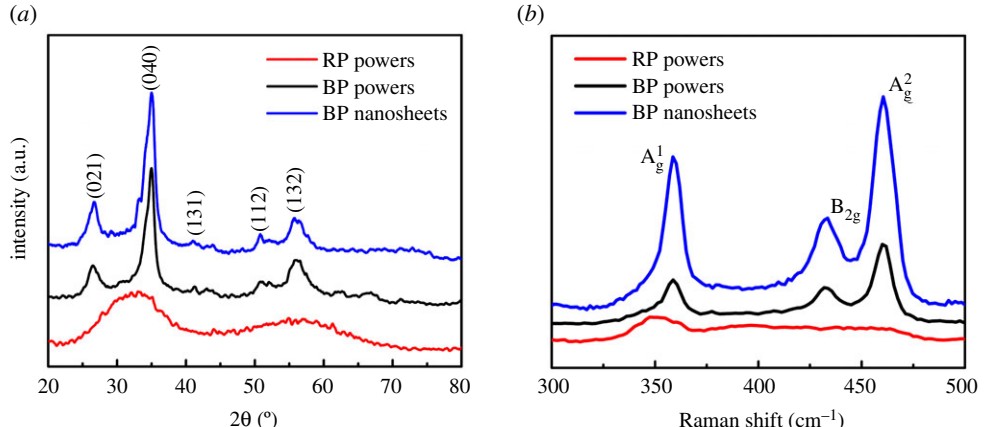

**Figure 2.** (*a*) XRD patterns of the RP and BP powders, (*b*) Raman spectra of the RP, BP powders and BP nanosheets.

The tribological properties of the LP and BP-LP lubricants were evaluated by a rotating ball-on-disc tribometer. The upper samples were the steel ball (GCr15) with a diameter of 6 mm and a hardness of HRC 61–66. The lower samples were the titanium alloy disc (Ti6Al4 V alloy) with a diameter of 25 mm and a thickness of 8 mm. The friction and wear tests were conducted at the rotary speed of 150 r min$^{-1}$ and the load range from 8 N to 15 N for 30 min at ambient temperature. Coefficient of friction (COFs) were recorded automatically by the tribometer.

The phase structures of RP, BP powders and BP nanosheets were investigated by the X-ray diffractometer (XRD) with the scan rate of 4° min$^{-1}$. A Raman spectrophotometer were used to identify the structural units in the RP, BP powders and BP nanosheets. The surface morphologies of BP powders and nanosheets were characterized by scanning electron microscopy (SEM) coupled with energy dispersive spectrometry (EDS). The size and distribution of BP nanosheets were characterized by transmission electron microscope (TEM). The morphologies and chemical compositions of the worn surfaces of the balls and discs were characterized by optional microscope (OM), SEM equipped with EDS, X-ray photoelectron spectroscopy (XPS).

# 3. Results and discussion

## 3.1. Characterization of black phosphorus

The XRD patterns and Raman spectroscopies of RP, BP powders and BP nanosheets are shown in figure 2*a*,*b*. It can be seen that RP has two large and broad diffraction peaks at 33° and 55°, indicating that RP has an amorphous structure. After HEBM for 2 h, two broader diffraction peaks of RP disappeared. The sharp diffraction peaks from 25° to 70° appeared and the main characteristic peaks at $2\theta = 25°$, 35° and 56° are presented. These results are consistent with standard orthorhombic BP (JCPDS no. 76–1957). It means that the phase transformation from RP to BP occurred during HEBM.

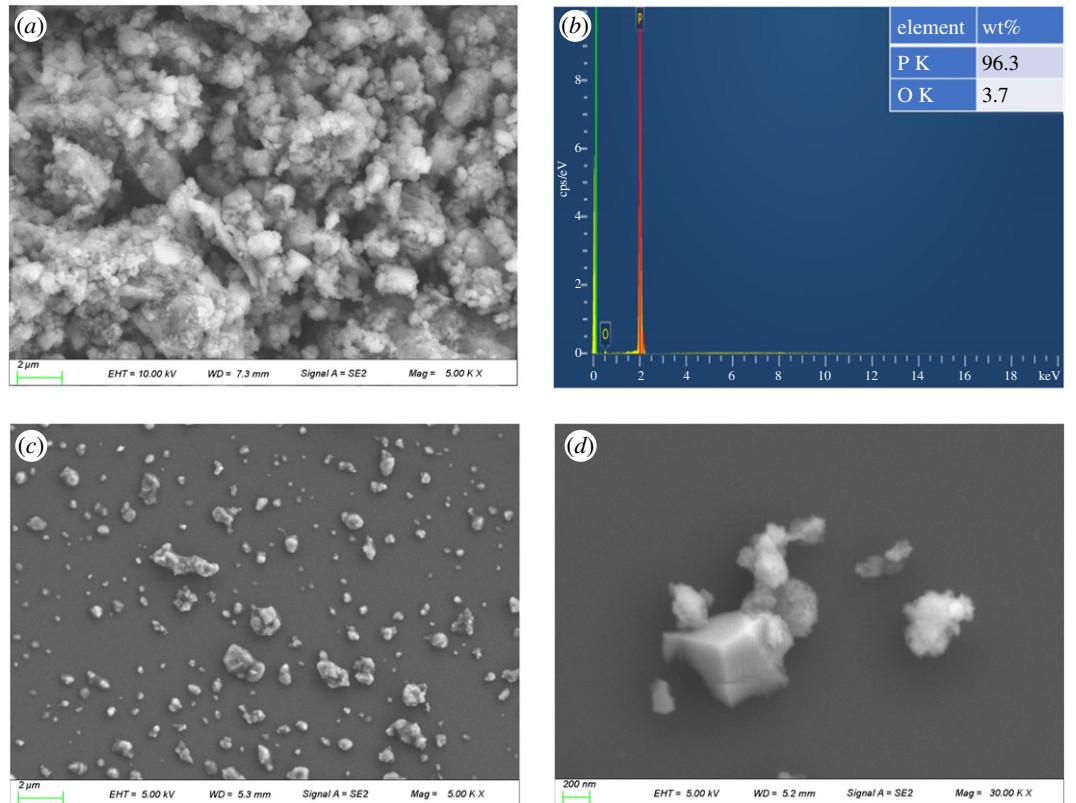

**Figure 3.** (*a*) SEM image and (*b*) EDS image of the BP powders (*c*), (*d*) SEM images of BP nanosheets.

Compared with RP, the X-ray diffraction peak intensities of BP powers were significantly enhanced. It indicates that the crystallinity of BP powers is also improved owing to the high temperature and pressure produced during HEBM. When the BP powers were exfoliated into nanosheets, the observed blue shift in the patterns happens. Furthermore, it was found that all RP powders have been transformed into BP powders after HEBM for 2 h. To further characterize the microstructure of RP and BP, Raman spectroscopies of RP and BP powers are presented in figure 2*b*. Raman characteristic peaks of RP powders at 350 cm$^{-1}$ (B1 basic mode), 394.6 cm$^{-1}$ (A1 symmetrical stretching vibration) and 461.6 cm$^{-1}$ (degenerate mode) are consistent with those reported in the literature [20–24]. After RP transformed into BP, all diffraction peaks of BP are sharper than that of RP, indicating that BP nanoparticles have better crystallinity. After HEBM, the Raman characteristic peaks of RP also changed correspondingly. The broad peak of RP near 350 cm$^{-1}$ disappeared, while three sharp diffraction peaks at 358.69 cm$^{-1}$, 432.88 cm$^{-1}$ and 460.01 cm$^{-1}$ in the spectrum of BP appeared, corresponding to the phonon modes $A_g^1$, $B_{2g}$ and $A_g^2$, respectively, and is in good agreement with the literature [25,26]. These corresponding peaks were attributed to the lattice vibration of BP crystal. The Raman spectrum of the liquid exfoliated BP nanosheets also exhibits the identical structural features seen at 359.01, 433.63 and 461.37 cm$^{-1}$ corresponding to the $A_g^1$, $B_{2g}$ and $A_g^2$ phonon modes, respectively. The blue shift phenomenon was found in the positions of Raman peaks along with small variation in their full width at half maximum values [25]. There is offset in the diffraction peaks of BP nanosheets, which is owing to the decreased thickness of BP nanosheets [27].

Figure 3 shows SEM and EDS images of the as-prepared BP powders and BP nanosheets. The microparticles were presented from 1 μm to 100 μm and concentrated at 2 μm. The EDS analysis indicates that the content of P is 96.3% (wt%) and the content of O is 3.7% (wt%). The SEM image of BP nanosheets (figure 3*c*) presents uniform distribution and small size, it clearly reveals sharp edges at higher magnification [28]. Because of poor dispersity of BP powders in the base oil, the BP nanosheets fabricated by liquid exfoliation were used in this investigation as BP-LP lubricants. BP nanosheets were further characterized by TEM and high resolution TEM (HRTEM) in figure 4. From figure 4*a*, it shows that the obtained BP nanosheets have a length of 200 nm and a width of 70 nm. From the HRTEM image (figure 4*b*) and corresponding selected area electron diffraction (SAED) pattern (figure 4*c*), it can be found that BP nanosheets used in this

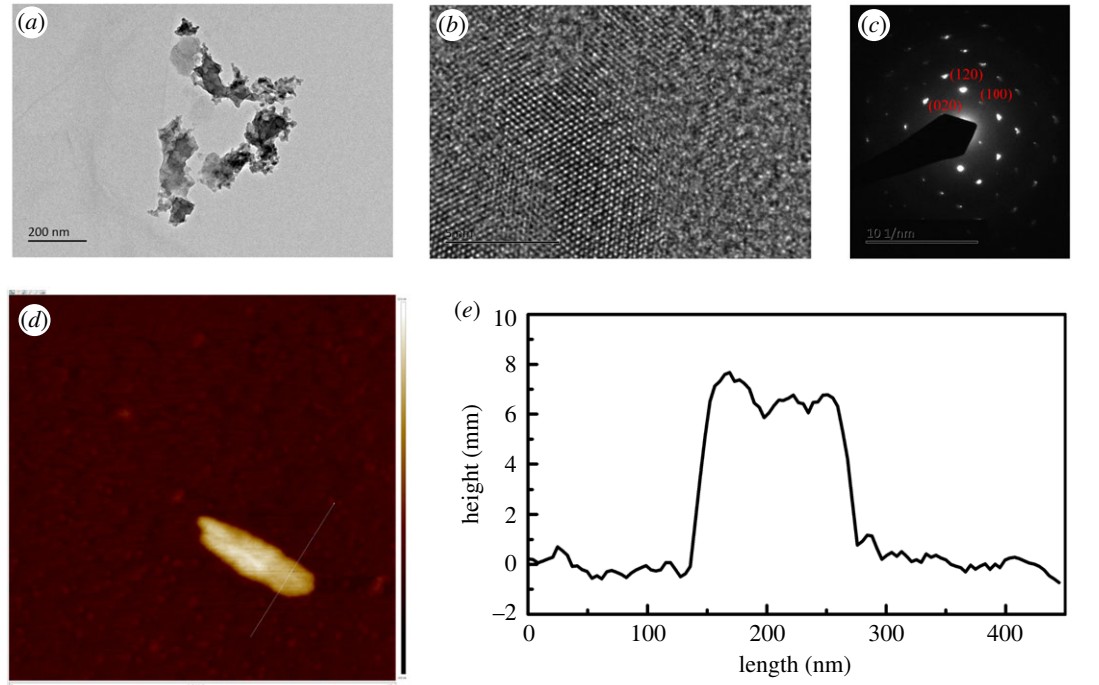

**Figure 4.** (*a*) Morphology, (*b*) HRTEM and (*c*) SAED of BP nanosheets in TEM image. (*d*) AFM image of BP nanosheets and (*e*) height profile corresponding to the AFM image.

investigation are orthorhombic in crystal structure and have high crystallinity [29,30]. Figure 4*d* shows the AFM image of BP nanosheets, the height of BP nanosheets (figure 4*e*) was measured at around 8 nm. It indicated that BP nanosheets have a multilayer structure, which is consistent with TEM image.

## 3.2. Tribological properties of black phosphorus oil-based lubricant additive

Figure 5 shows COFs of LP and BP-LP lubricants at the loads of 8 N, 10 N, 12 N and 15 N (corresponding to 1039 MPa, 1119 MPa, 1190 MPa, 1281 MPa). It was clearly found that COFs of LP are higher than that of BP-LP at the same load. At the load of 8 N, COF of LP was increased to 0.38 at the beginning, and then drastically dropped to 0.31, and finally fluctuated between 0.30 and 0.32. Compared with COFs of LP, COFs of BP-LP were significantly decreased. As the load increased to 12 N, COFs of LP were unstable in the first 10 min. COFs of BP-LP were maintained at 0.26, the friction coefficient curve tended to be smooth. When the load increased to 15 N, COFs of LP remained at about 0.26, while COFs of BP-LP were basically stabilized around 0.23 after 5 min. In general, the friction coefficient curves of BP-LP were always lower than that of LP at the same load.

In order to study the evolution of the COFs under different loads, the average COFs and wear rates of GCr15 steel balls are shown in figure 6. It can be seen that as the load increased, the average COFs and wear rates of LP and BP-LPv presented similar variations. As the load increased from 8 N to 10 N, the average COFs and wear rates of LP and BP-LP were increased, while as the load increased from 10 N to 15 N, the average COFs and wear rates of LP and BP-LP were all decreased. At the same loads, the average COFs and wear rates of BP-LP are lower than that of LP. At the load of 8 N, although the average COF of BP-LP was reduced by 10% compared with the average COF of LP, the wear rate of BP-LP was reduced by 45.2%. Similarly, the COF of BP-LP reached 0.237 at 15 N, which was reduced by 30%, and the wear rate at this load reached a minimum value of $0.79 \times 10^{-9}$ mm N$^{-1}$, which showed the best anti-friction performance.

The above phenomenon can be explained by Hertz's theory [31]. The tribological properties of BP-LP lubricants were investigated by sphere-on-disc friction tests. Sphere contact as object and the contact stress formula of Hertz are deduced theoretically from the contact model of spheres and disc [32], as shown in figure 7. When the sphere is in contact with the disc under the external loading, elliptic contact area will be produced around the contact point owing to the partial deformation of sphere and disc.

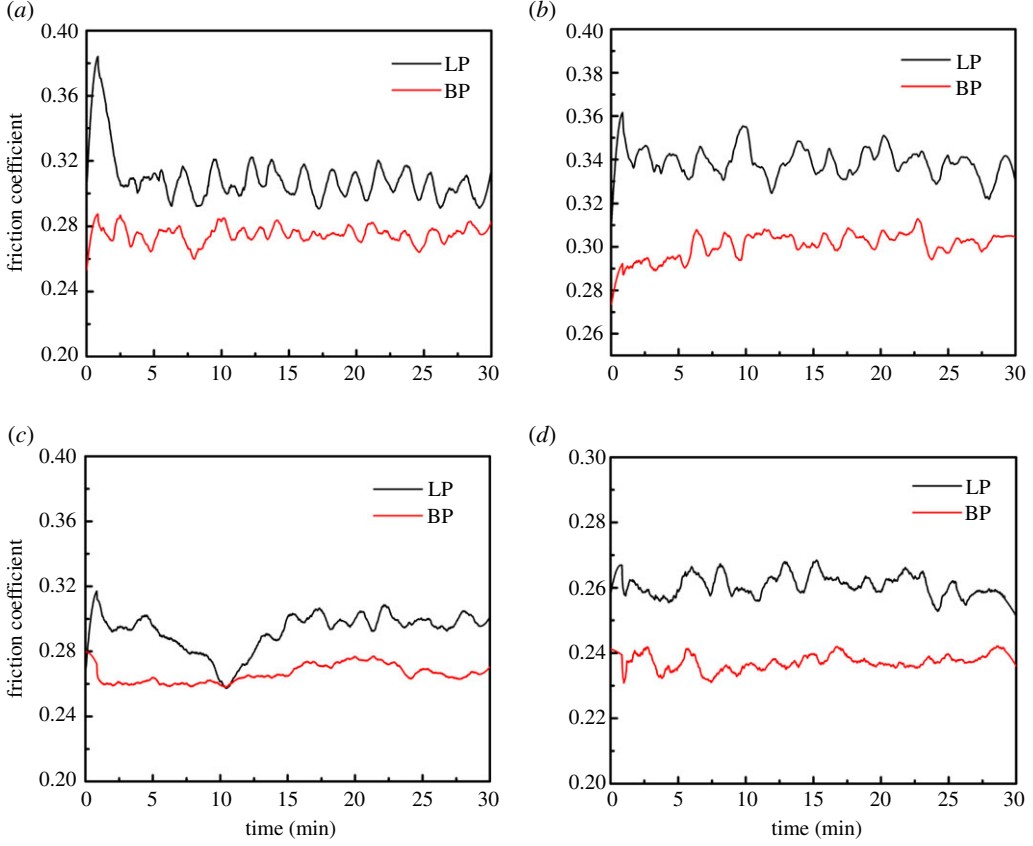

**Figure 5.** The friction coefficient curve of LP and BP-LP under different loads: (*a*) 8 N, (*b*) 10 N, (*c*) 12 N, and (*d*) 15 N.

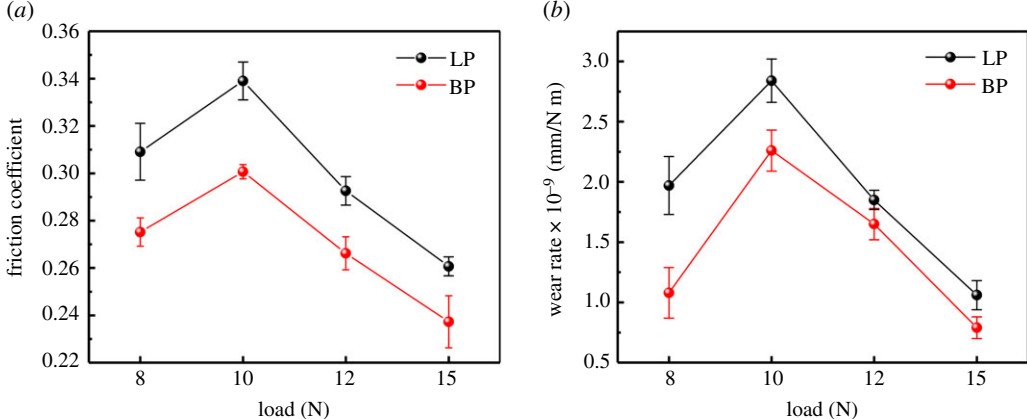

**Figure 6.** (*a*) Friction coefficients and (*b*) wear rates of LP and BP-LP under various load conditions.

The corresponding contact stresses were calculated according to equation (3.1):

$$q_0 = \frac{4F}{\pi a^2} \tag{3.1}$$

and

$$a = 2\left(\frac{2}{3} * \frac{FR}{E'}\right), \tag{3.2}$$

where, $q_0$ and $\alpha$ are maximum Hertz contact stress and Hertz contact diameter, respectively, $F$ is the normal load (8 N, 10 N, 12 N, 15 N), $R$ is the radius of the ball (3 mm), and $E'$ is the effective elastic

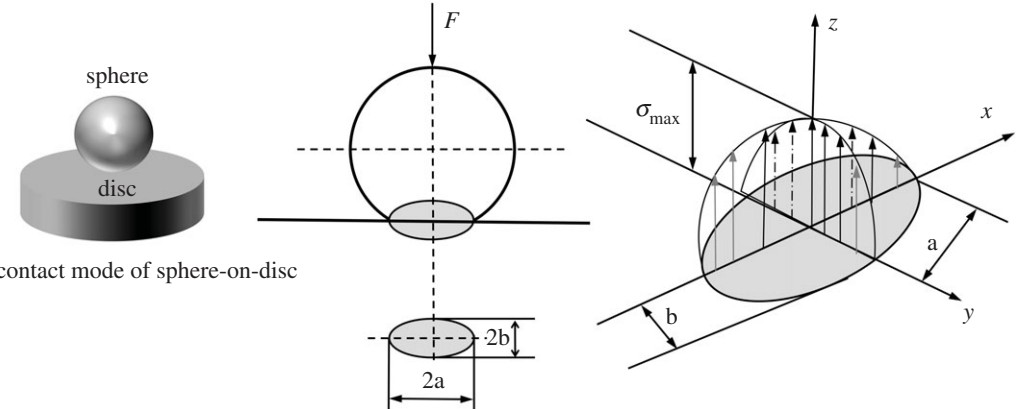

**Figure 7.** Hertz contact model of sphere-on-disc.

modulus. It can be expressed by the following formula [33]:

$$\frac{1}{E'} = \frac{1}{2}\left(\frac{1-\mu_1^2}{E_1} + \frac{1-\mu_2^2}{E_2}\right),\tag{3.3}$$

where $E_1$ (110 GPa) and $E_2$ (210 GPa) are the elastic moduli of the TC4 and GCr15, respectively, and $\mu_1$ (0.34) and $\mu_2$ (0.30) are the Poisson ratios of the TC4 and GCr15, respectively. Thus, the corresponding maximum Hertzian contact stresses are 1039 MPa, 1119 MPa, 1190 MPa and 1281 MPa. As for equation (3.2), the elliptic contact area is increased proportional to $F$, which indicates that the contact area increased more slowly than that of load. Based on the formulae of the friction coefficient, $\mu = f/F$ ($\mu$ is the COF, $f$ is the frictional force, $F$ is the load), it can be concluded that the values of the COF may be reduced as the load increased. The wear volumes of the ball side are calculated by the following formula [34]:

$$V = \left(\frac{\pi l}{6}\right)\left(\frac{3d^2}{4} + l^2\right)\tag{3.4}$$

and

$$l = r - \sqrt{r^2 - \frac{d^2}{4}},\tag{3.5}$$

$V$ is the wear volume, $r$ is the radius of the ball, $d$ is the wear scar diameter.

Hence, the wear rate was decreased as the load increased and reduced the wear. These results showed that adding BP nanosheets as a lubrication additive into LP can significantly improve the lubricating properties of the base oil of LP and effectively reduce the wear of the GCr15/TC4 friction pair.

## 3.3. Worn surface analysis

The OM images of the wear scars on the GCr15 ball surface lubricated by LP and BP-LP under various load conditions are shown in figure 8. From the wear scars of the ball, it can be seen that the appearance of all the wear scars is elliptic. The main reason for this is that when the sphere is in contact with the disc under the external loading, the elliptic contact area will be produced around the contact point owing to the partial deformation of sphere and disc. When the GCr15 ball comes into contact with the disc under the external force of the load, an elliptical contact area around the contact point was formed owing to the local deformation of the ball and the disc [32]. It can be observed that the surface of the steel ball lubricated by LP has deep and wide abrasion marks along the sliding direction, which is rather rough and uneven, and there are obvious scratches and pits, and the wear volume is larger. On the contrary, after adding the BP nanosheets as lubricant additives into LP, the wear scars are shallow and narrow, the friction surface becomes more uniform. There are slight scratches and the wear volume becomes smaller, which is consistent with the calculation results of the wear rates. Especially, the wear volume of balls lubricated by BP-LP are the smallest and narrowest at the load of 15 N. These results showed that the BP nanosheets could effectively improve the wear resistance of the base oil of LP. Figure 9 shows the SEM images of the wear scar of TC4 disc lubricated by LP and BP-LP at different loads.

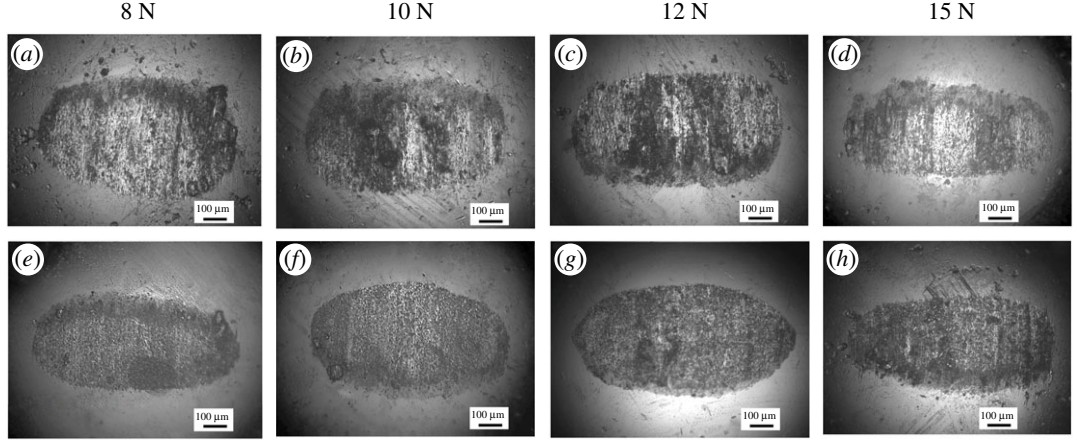

**Figure 8.** OM images of wear scars on the GCr15 surface lubricated by different lubricants under different loads, (*a–d*) LP lubricants, (*e–h*) BP-LP lubricants.

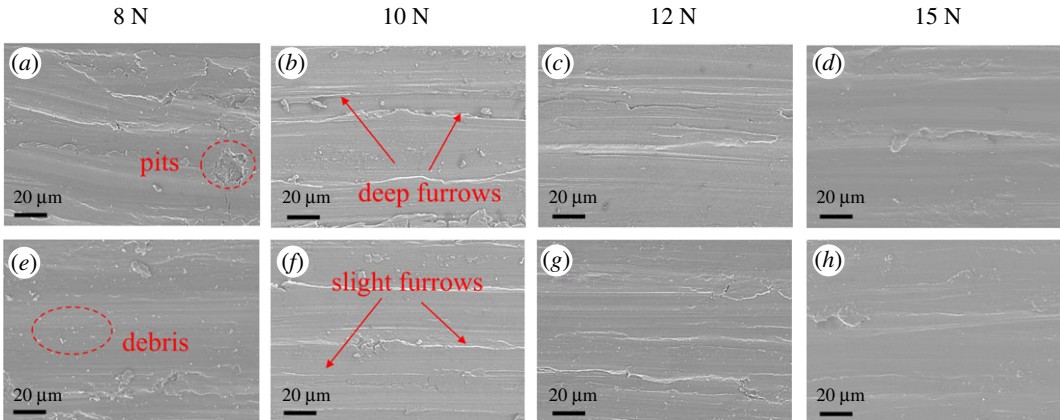

**Figure 9.** The SEM images of the wear scar of the TC4 disc lubricated by different lubricants under different loads: (*a–d*) LP lubricants, (*e–h*) BP-LP lubricants.

For the LP lubricants, there are many large-scale spalling pits on the surface of the wear scars. It presents the abrasive wear and fatigue wear characteristics. The deep furrows were mainly concentrated in the middle of the wear scar, indicating that the central region is the serious wear region. Figure 9*e–h* are the worn surfaces of BP-LP lubricants. It can be seen that some scattered debris and the furrow wear characteristics were presented on the surface of the TC4 disc. Compared with the wear surface of LP, the BP-LP lubricants have smoother wear surfaces at corresponding loads. It indicates that the addition of BP nanosheets enhance the wear resistance of the TC4 disc.

Figure 10 shows the EDS spectra of the wear scar of the TC4 disc lubricated by the base oil of LP. As it can be seen in figure 10*a*, there are obvious desquamations on the wear surface. Except for the base elements of the TC4 matrix, a large amount of C elements were enriched on the wear surface. This is mainly derived from the base oil of LP. In addition, it can be noted that the Fe elements were also presented on the wear surface. The reason for this is that during the sliding process the Fe elements in the upper sample of GCr15 were transferred into the surface of the lower sample of the TC4 alloy. It indicated that adhesion wear occurred during the friction process. From the EDS spectrum of the wear surface lubricated by BP-LP lubricants in figure 11, after adding BP nanosheets in the base oil of LP, the surface of the wear scar is obviously smooth and the furrow becomes shallow. From figure 11*b–d*, it can be seen that the chemical compositions of the wear surface mainly contained five elements including P, C, Ti, V and Al. These elements are mainly coming from the TC4 matrix and the base oil of LP. In addition, a small amount of P elements was also observed in the wear surface, it has derived from the BP nanosheets. These results further suggested that the lamellar BP nanosheets are squeezed to the contact area between the ball and the disc owing to frictional stress. The BP nanosheets were adsorbed on the surface of the TC4 disc, thereby it could reduce the friction effect of

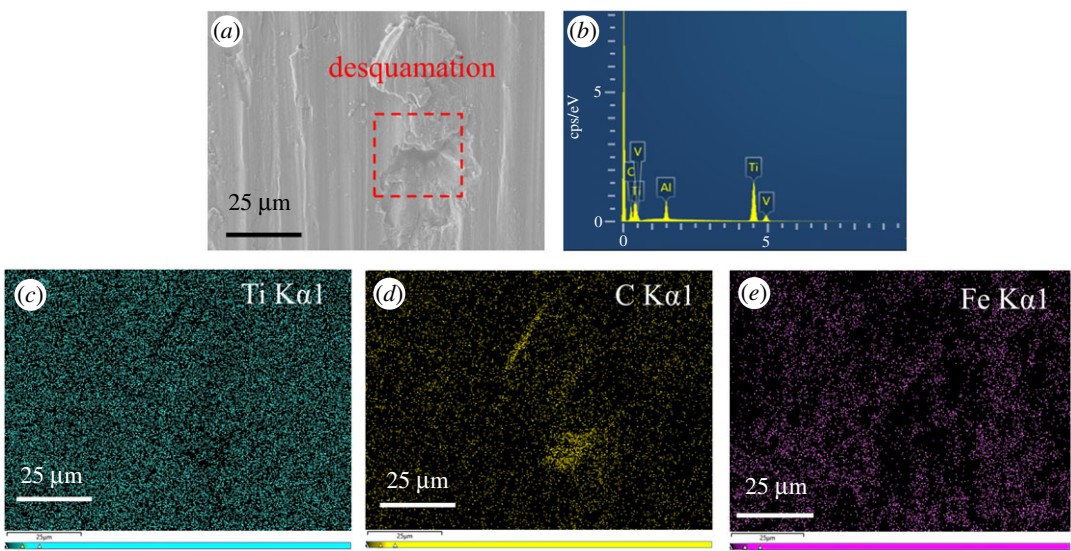

**Figure 10.** The EDS spectra of the wear scar of the TC4 disc lubricated by LP. (*a*) The high resolution SEM image, (*b*) the spectra, (*c–e*) the distribution of Ti, C, Fe elements.

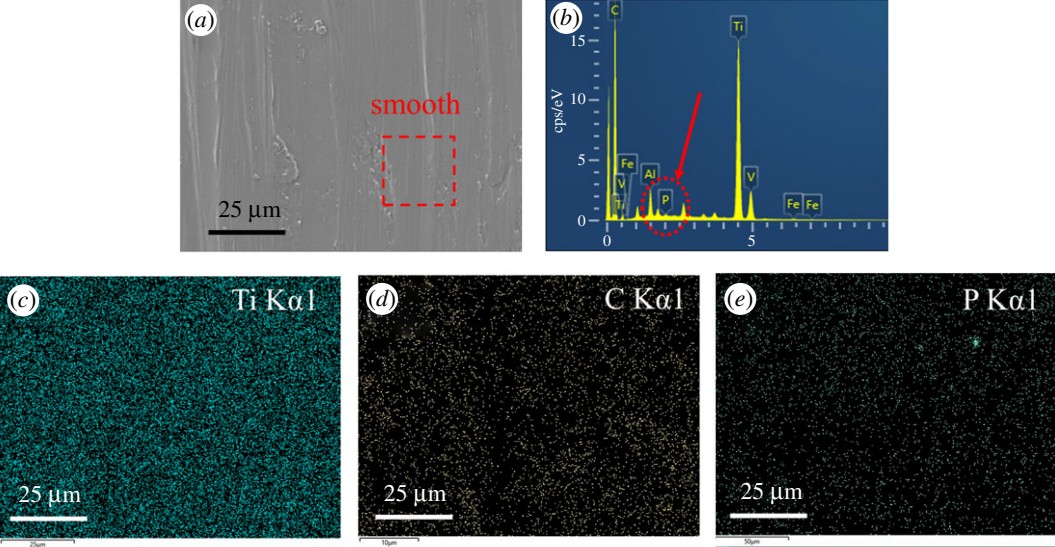

**Figure 11.** The EDS spectra of the wear scar of the TC4 disc lubricated by BP-LP. (*a*) The high resolution SEM image, (*b*) the spectra, (*c–e*) the distribution of Ti, C, P elements.

the friction pairs on the substrate and improve its abrasion resistance, indicating that the abrasion resistance of the TC4 disc is significantly improved.

In order to further explain the lubrication mechanism of BP-LP lubricants, the curve-fitted XPS spectra of C1s, O1s and P2p on the wear scar of the disc lubricated with BP-LP are shown in figure 12. The C1's peak at 285.04 eV is attributed to the C–C or C–H bond that comes from the organics in LP [35], and the peak at 284.64 eV is attributed to the C–C or C=C bond. These results indicate that LP molecules are adsorbed on the worn surface owing to its viscosity. Therefore, the physical adsorption film was formed on the surface of the steel ball. In the oxygen region, the O1's peak at 531.73 eV is attributed to the C–O or C=O bond [36]. With the addition of BP nanosheets into the base oil, the weak P2p peak around 132.97 eV is identified as $PO_4^{3-}$ [14]. It is worth mentioning that phosphorus may have been oxidized in the friction process, it means that the formed protective tribofilm can prevent the direct contact of friction pairs. According to classical lubrication theories, the lubrication regimes can be identified in a lubrication regime map in terms of the two variables ($g_v$ and $g_E$):

$$g_v = \frac{GW^3}{U^2} \tag{3.6}$$

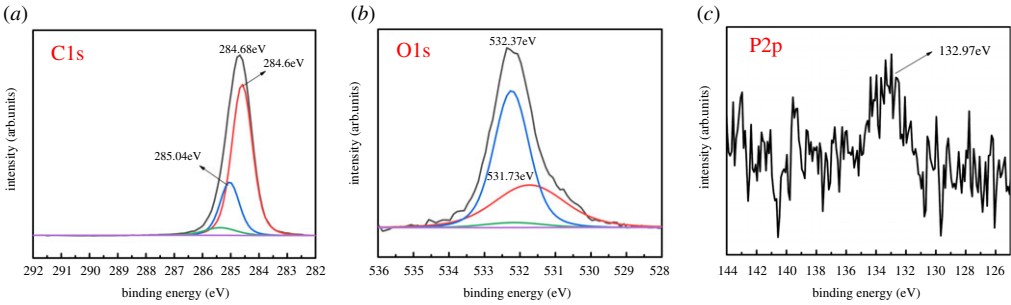

**Figure 12.** XPS analysis of the worn surfaces on the TC4 disc lubricated by BP-LP: (a–c) corresponding XPS spectra of C, O and P obtained from the worn surface.

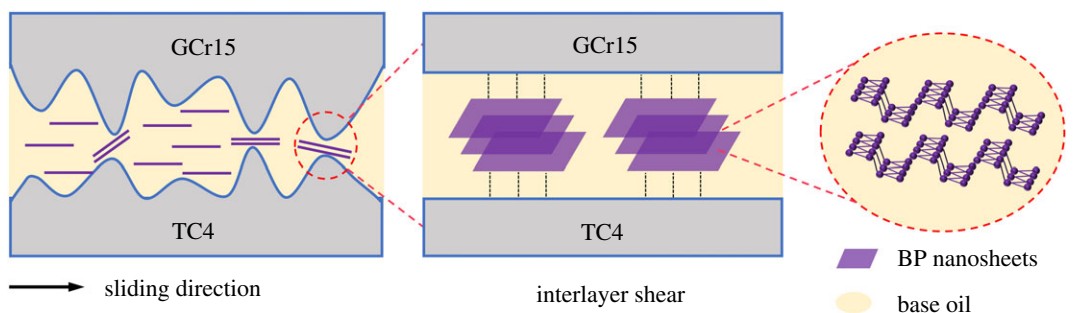

**Figure 13.** Schematic diagram of lubrication mechanism of BP-LP.

and

$$g_{\mathrm{E}} = \frac{W^{8/3}}{U^2}, \tag{3.7}$$

where $U = \eta V / E'R$, $G = \alpha E'$, $W = F / E'R^2$, $V$ is the averaged linear velocity (62.8 mm s$^{-1}$), $\eta$ is the bulk viscosity of lubricating solution, $\alpha$ is the viscosity-pressure coefficient, $R$ is the radius of the ball (3 mm), $E'$ is the effective modulus of the elasticity (162 GPa), $F$ is the normal load (15 N), $k$ ($\approx$1.03) is the elliptical parameter. According to Hamrock-Dowson theory, the minimum film thickness can be calculated using following formula [37]:

$$h_{\min} = 2.69 \frac{G^{0.53} U^{0.67}}{W^{0.067}} (1 - 0.61 \mathrm{e}^{-0.73k}). \tag{3.8}$$

The lubrication regime can be determined by the ratio of theoretical minimum film thickness to the combined surface roughness. The following formula can be used to calculate the ratio $\gamma$:

$$\gamma = \frac{h_{\min}}{\sqrt{\sigma_1^2 + \sigma_2^2}}, \tag{3.9}$$

where $\sigma_1$ and $\sigma_2$ are the roughness of the opposite rubbing surfaces after the lubrication tests ($\sigma_1 = 40$ nm, $\sigma_2 = 20$ nm).

Finally, the calculated $h_{\min}$ is about 7.54 nm when the load is 15 N and the speed is 62.8 mm s$^{-1}$. Thus, the calculated ratio of film thickness to surface roughness $\lambda$ is approximately 0.16, is smaller than 1, and thereby indicating the current lubrication is in the regime of boundary lubrication [38]. Therefore, it can improve the tribological performance of LP and it can be concluded that the electrostatic adsorption and interlayer shear on the interface. BP nanosheets play an important role in the friction-reducing ability of BP-LP lubricants.

Based on the above test results and discussions, figure 13 shows the schematic diagram of the lubrication mechanism of BP-LP. It is inferred that the excellent tribological properties of the BP nanosheets as lubricant additives are mainly owing to the following two aspects. On one hand, the base oil has its own viscosity during the initial friction. Owing to the high surface activity, the BP nanosheets dispersed in LP can be deposited quickly on the surface of the TC4 disc through physical

adsorption [32], and form a physical lubricating film. Thus, it could avoid the direct contact of the friction pair and reduce wear and friction. The BP nanosheets are deposited on the surface of the friction pair through physical adsorption, preventing direct contact of rough peaks between the friction pairs, and improving the antifriction performance of the base oil. On the other hand, as the loads increased, the lubricating film will be damaged by external extrusion, friction and heat during the friction process. The low shear forces between the rough peaks during the sliding process were formed owing to the lamellar structure of BP nanosheets and the combination of Van Der Waals forces. BP nanosheets were diffused into the grooves in the contact area of friction process owing to the electrostatic adsorption. It could achieve better friction-reducing and wear resistance. Therefore, lubrication mechanisms of BP-LP are the synergistic lubrication between lamellar adsorption and the interlayer shear of BP nanosheets.

# 4. Conclusion

In this investigation, BP nanosheets as the LP-based lubrication additive were prepared by high energy ball milling, liquid-phase exfoliation and the ultrasonic dispersion method. The tribological properties of the base oil of LP and BP-LP lubricants were researched at different loads. Compared with LP, the COF of the BP lubrication additive can be reduced by 30% and the wear rate was reduced by 45.2%. The results indicated that the addition of BP nanosheets in the base oil can significantly improve the antifriction and wear resistance of LP for GCr15-TC4 contacts. The lubrication mechanisms of BP-16C are mainly owing to the synergistic effect of lamellar adsorption and interlaminar shearing of BP nanosheets, which have good tribological properties as an oil-based lubrication additive.

Data accessibility. This article does not contain any additional data.

Authors' contributions. Q.W., T.H. and W.W. designed the investigation. Q.W. prepared all samples for analysis. T.H. and W.W. collected and analysed the data. G.Z., Y.G. and K.W. performed statistical analysis and collected field data. T.H. and W.W. interpreted the results and wrote the manuscript.

Competing interests. We declare we have no competing interests.

Funding. Financial support came from the Research Fund of the Guangdong Science and Technology Program (2017B030314002), the National Natural Science Foundation of China (grant no. 51605249, 51975450), International Scientific and Technological Cooperation Program of the Shaanxi Province (grant nos. 2019KW-026, 2019KW-064).

Acknowledgements. We thank instructional support specialists of Xi'an University of Architecture and Technology laboratory. We also grateful to anonymous reviewers, who provided comments that improved the manuscript. We guarantee that this thesis is an original work, without multiple submissions, and does not involve confidentiality and other copyright-related infringement issues. In the event of multiple submissions, infringements, leaks, etc. we are responsible for all responsibilities.

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
