## [Reviewer comments · Royal Society Open Science]

Review History

RSOS-200530.R0 (Original submission)

Review form: Reviewer 1

Is the manuscript scientifically sound in its present form?

Yes

Are the interpretations and conclusions justified by the results?

Yes

Is the language acceptable?

Yes

Do you have any ethical concerns with this paper?

No

Have you any concerns about statistical analyses in this paper?

No

Recommendation?

Accept with minor revision (please list in comments)

Comments to the Author(s)

The PDF file has been attached (Appendix A).

Review form: Reviewer 2 (Shih-Chen Shi)

Is the manuscript scientifically sound in its present form?

Yes

Are the interpretations and conclusions justified by the results?

Yes

Is the language acceptable?

Yes

Do you have any ethical concerns with this paper?

Yes

Have you any concerns about statistical analyses in this paper?

No

Recommendation?

Accept with minor revision (please list in comments)

Comments to the Author(s)

- 1) The discussion part is not strong enough, most of the articles are explaining the data. It's really difficult to persuade the author's point of view, especially for the lubricating mechanism.
- 2) It is recommended to calculate wear from the profile of the wear scar, so that the mechanism of wear can be more accurately observed.

Decision letter (RSOS-200530.R0)

Dear Professor Wang:

Title: Tribological properties of black phosphene nanosheets as oil-based lubricant additives for titanium alloy-steel contacts

Manuscript ID: RSOS-200530

Thank you for submitting the above manuscript to Royal Society Open Science. On behalf of the Editors and the Royal Society of Chemistry, I am pleased to inform you that your manuscript will be accepted for publication in Royal Society Open Science subject to minor revision in accordance with the referee suggestions. Please find the reviewers' comments at the end of this email.

The reviewers and handling editors have recommended publication, but also suggest some minor revisions to your manuscript. Therefore, I invite you to respond to the comments and revise your manuscript.

Because the schedule for publication is very tight, it is a condition of publication that you submit the revised version of your manuscript before 11-Jul-2020. Please note that the revision deadline will expire at 00.00am on this date. If you do not think you will be able to meet this date please let me know immediately.

Kind regards,
Dr Laura Smith
Publishing Editor, Journals

On behalf of the Subject Editor Professor Anthony Stace and the Associate Editor Dr Dattatray Late.

RSC Subject Editor:
Comments to the Author:
(There are no comments.)

RSC Associate Editor:
Comments to the Author:
NA

Reviewer comments to Author:
Reviewer: 1

Comments to the Author(s)
The PDF file has been attached

Reviewer: 2

Comments to the Author(s)
1) The discussion part is not strong enough, most of the articles are explaining the data. It's really difficult to persuade the author's point of view, especially for the lubricating mechanism.
2) It is recommended to calculate wear from the profile of the wear scar, so that the mechanism of wear can be more accurately observed.

Author's Response to Decision Letter for (RSOS-200530.R0)

See Appendix B.

RSOS-200530.R1 (Revision)

Review form: Reviewer 1

Is the manuscript scientifically sound in its present form?

Yes

Are the interpretations and conclusions justified by the results?

Yes

Is the language acceptable?

Yes

Do you have any ethical concerns with this paper?

No

Have you any concerns about statistical analyses in this paper?

No

Recommendation?

Accept as is

Comments to the Author(s)

The authors have carefully considered all suggestions and criticisms of both referees. I find the answers to my questions and suggestions very convincing. The present manuscript is highly likely to initiate interesting discussions and follow-up studies. I strongly recommend publication of the manuscript in its present form.

Decision letter (RSOS-200530.R1)

Dear Professor Wang:

Title: Tribological properties of black phosphene nanosheets as oil-based lubricant additives for titanium alloy-steel contacts
Manuscript ID: RSOS-200530.R1

It is a pleasure to accept your manuscript in its current form for publication in Royal Society Open Science. The chemistry content of Royal Society Open Science is published in collaboration with the Royal Society of Chemistry.

On behalf of the Subject Editor Professor Anthony Stace and the Associate Editor Dr Dattatray Late.

RSC Associate Editor:
Comments to the Author:
(There are no comments.)

RSC Subject Editor:
Comments to the Author:
(There are no comments.)

Reviewer(s)' Comments to Author:
Reviewer: 1

Comments to the Author(s)

The authors have carefully considered all suggestions and criticisms of both referees. I find the answers to my questions and suggestions very convincing. The present manuscript is highly likely to initiate interesting discussions and follow-up studies. I strongly recommend publication of the manuscript in its present form.

Appendix A

Manuscript title: Tribological properties of black phosphene nanosheets as oil-based lubricant additives for titanium alloy-steel contacts.

Manuscript ID: RSOS-200530

The authors rigorously studied the black phosphorus (BP) powders preparation by high-energy ball milling with red phosphorus as source material. Further BP nanosheets obtained by liquid-phase exfoliation and studied oil-based lubricant additives of synthesised materials. The paper is rich in data, novelty, used versatile characterization tool, explain band alignment and the general conclusion of the paper sounds. These facts make the paper suitable to be published in Royal Society Open Science, in principle. However, some minor corrections must be done before its acceptance.

- 1) I am not satisfied with the assigned XRD planes of BP, can you please make comment on it. You can see the literature the peak $\sim 34\text{-}35$ degree is always assigned as $\langle 040 \rangle$ plane and same for others planes.
- 2) In figure 2, I strongly recommend to add XRD pattern and Raman spectra of (i) red phosphorus (ii) black phosphorus powder and (iii) black phosphorus nanosheets obtained by liquid-phase exfoliation
- 3) Assign the vibrational mode “ B_{2g} ” which is appeared $\sim 438\text{ cm}^{-1}$ according to RSC Adv., 2016, 6, 76551–76555, ACS Appl. Mater. Interfaces 2015, 7, 5857–5862 and ACS Appl. Mater. Interfaces 2016, 8, 11548–11556 references.
- 4) Add note in the manuscript about position change of raman peak and xrd angle for BP powder and nanosheets.
- 5) What is the average thickness of BP nanosheets ? please add relevant characterisation.

- 6) Can you please add high magnification SEM/TEM with same scale bar image of BP nanosheets and powder in figure 3.
- 7) Authors are encouraged to cite the following 6 articles in the introduction, Raman , XRD and TEM characterization, who did an extensive study on black phosphorus nanosheets:
(i) ACS Applied Materials & Interfaces 8 (18), 11548–11556 (2016) **(ii)** RSC advances 6 (80), 76551-76555 (2016) **(iii)** Applied Physics A 124 (2), 133 (2018) **(iv)** ACS Applied Materials & Interfaces 7 (10), 5857–5862 (2015) **(v)** Microporous and Mesoporous Materials 225, 494–503 (2016) **(vi)** RSC Advances 6 (113), 112103-112108 (2016)

In summary, I believe this is an interesting study, and the results could potentially be interesting and useful for 2D community and Materials Science. However, the above comments should be well addressed before the paper can be considered for publication.

Appendix B

Resubmission of RSOS-200530

Dear Dr Laura Smith,

We would like to express our gratitude to your efforts devoted to evaluating our manuscript entitled “*Tribological properties of black phosphene nanosheets as oil-based lubricant additives for titanium alloy-steel contacts*”, which have been submitted to RSOS for the first-peer review.

We appreciate the reviewer’s constructive comments on our manuscript and your kind consideration of publishing this manuscript in Royal Society Open Science. We have addressed all of the comments and revised the manuscript accordingly. All of the changes have been highlighted by red font in the marked-up manuscript. Details of our replies to the comments and the revisions are described in the “Detailed Responses to the Reviewer Comments”.

We now resubmit the manuscript for your further consideration for publication in Royal Society Open Science.

Sincerely yours,

Dr. Wei Wang.

Detailed Responses to the Reviewer Comments-1

The authors rigorously studied the black phosphorus (BP) powders preparation by high-energy ball milling with red phosphorus as source material. Further BP nanosheets obtained by liquid phase exfoliation and studied oil-based lubricant additives of synthesized materials. The paper is rich in data, novelty, used versatile characterization tool, explain band alignment and the general conclusion of the paper sounds. These facts make the paper suitable to be published in Royal Society Open Science, in principle. However, some minor corrections must be done before its acceptance.

1) I am not satisfied with the assigned XRD planes of BP, can you please make comment on it. You can see the literature the peak $\sim 34\text{-}35$ degree is always assigned as $\langle 040 \rangle$ plane and same for others planes.

Response: Thank you for your suggestion. We have revised the assigned XRD planes of BP, the peak $\sim 34\text{-}35$ degree was assigned as $\langle 040 \rangle$ in the figure 2. The revised Fig.2 was presented in the following parts.

The XRD patterns and Raman spectroscopies of RP, BP powders and BP nanosheets are shown in Fig. 2(a) and (b). It can be seen that RP has two large and broad diffraction peaks at 33° and 55° , indicating that RP has an amorphous structure. After HEBM for 2h, two broader diffraction peaks of RP were disappeared. The sharp diffraction peaks from 25° to 70° were appeared and the main characteristic peaks at $2\theta = 25^\circ$, 35° and 56° were presented. These results are consistent with standard orthorhombic BP (JCPDS No. 76-1957). It means that the phase transformation from RP to BP was occurred during HEBM. Compared with RP, the X-ray diffraction peak intensities of BP powers were significantly enhanced. It indicates that the crystallinity of BP powers is also improved owing to the high temperature and pressure produced during HEBM.

Fig. 2 (a) XRD patterns of the RP and BP powders, (b) Raman spectra of the RP, BP powders and BP nanosheets

2) In figure 2, I strongly recommend to add XRD pattern and Raman spectra of (i) red phosphorus (ii) black phosphorus powder and (iii) black phosphorus nanosheets obtained by liquid-phase exfoliation.

Response: Thank you for your suggestion. We have added XRD pattern and Raman spectra of red phosphorus (ii) black phosphorus powder and (iii) black phosphorus nanosheets obtained by liquid-phase exfoliation in Fig.2, and give the corresponding explanation. The revised parts are marked the red front.

After HEBM, the Raman characteristic peaks of RP were also changed correspondingly. The broad peak of RP near 350 cm⁻¹ was disappeared, while three sharp diffraction peaks at 357.4 cm⁻¹, 438cm⁻¹ and 460 cm⁻¹ in the spectrum of BP were appeared [25,26]. These corresponding peaks were attributed to the lattice vibration of BP crystal. Besides, there are offset in the diffraction peaks of BP nanosheets, which is due to the decreased thickness of BP nanosheets [27].

3) Assign the vibrational mode “B2g” which is appeared ~ 438 cm⁻¹ according to RSC Adv.,2016, 6, 76551–76555, ACS Appl. Mater. Interfaces 2015, 7, 5857–5862 and ACS Appl.Mater. Interfaces 2016, 8, 11548–11556 references.

Response: Thank you for your suggestion. We have assigned the vibrational mode “B2g” in figure2(b), and added the references [25, 26, 27] in the manuscript. The revised parts are marked the red front.

[25] Pawbake A S, Erande M B, Jadkar S R, et al. Temperature dependent Raman spectroscopy of electrochemically exfoliated few layer black phosphorus

nanosheets[J]. RSC advances, 2016, 6(80): 76551-76555.

[26] Late D J. Temperature dependent phonon shifts in few-layer black phosphorus[J].

ACS applied materials & interfaces, 2015, 7(10): 5857-5862.

[27] Erande M B, Pawar M S, Late D J. Humidity sensing and photodetection behavior of electrochemically exfoliated atomically thin-layered black phosphorus nanosheets[J]. ACS applied materials & interfaces, 2016, 8(18): 11548-11556.

4) Add note in the manuscript about position change of raman peak and xrd angle for BP powder and nanosheets.

Response: Thank you for your suggestion. We have added some description in Page 6, line 9 and line 20. The revised parts are marked the red front. The revised parts are as follows.

The broad peak of RP near 350 cm^{-1} was disappeared, while three sharp diffraction peaks at 358.69 cm^{-1} , 432.88 cm^{-1} and 460.01 cm^{-1} in the spectrum of BP were appeared corresponding to the phonon modes A_g^1 , B_{2g} and A_g^2 , respectively, and is in good agreement with literature [25, 26]. These corresponding peaks were attributed to the lattice vibration of BP crystal. The Raman spectrum of the liquid exfoliated BP nanosheets also exhibits the identical structural features seen at 359.01 , 433.63 and 461.37 cm^{-1} corresponding to the A_g^1 , B_{2g} and A_g^2 phonon modes, respectively. The blue shift phenomenon was found in the positions of Raman peaks along with small variation in their full width at half maximum (FWHM) values [25]. There are offset in the diffraction peaks of BP nanosheets, which is due to the decreased thickness of BP nanosheets [27].

5) What is the average thickness of BP nanosheets? Please add relevant characterisation

Response: Thank you for your suggestion. We have added the AFM image of BP nanosheets in Fig.4(d), (e). It confirms that the average thickness of BP nanosheets was measured at around 8nm. The revised Fig.4 (d) and (e) are presented in the following parts.

Fig. 4 (a) Morphology, (b) HRTEM and (c) SAED of BP nanosheets in TEM image (d) AFM image of BP nanosheets and (e) height profile corresponding to the AFM image

6) Can you please add high magnification SEM/TEM with same scale bar image of BP nanosheets and powder in figure 3?

Response: Thank you for your suggestion. We have added the high magnification SEM image of BP nanosheets and powders in figure 3.

7) Authors are encouraged to cite the following 6 articles in the introduction, Raman , XRD and TEM characterization, who did an extensive study on black phosphorus nanosheets: ACS Applied Materials & Interfaces 8 (18), 11548–11556 (2016) (ii) RSC advances 6 (80), 76551-76555 (2016) (iii) Applied Physics A 124 (2), 133 (2018) (iv) ACS Applied Materials & Interfaces 7 (10), 5857–5862 (2015) (v) Microporous and Mesoporous Materials 225, 494–503 (2016) (vi) RSC Advances 6 (113), 112103-112108 (2016)

Response: Thank you for your suggestion. We have added the references [25, 26, 27] for the Raman and XRD characterization, added the references [28, 29, 30] for the TEM characterization in the manuscript. The revised parts are marked the red front. The added references are presented in the following parts.

[25] Pawbake A S, Erande M B, Jadkar S R, et al. Temperature dependent Raman

- spectroscopy of electrochemically exfoliated few layer black phosphorus nanosheets[J]. RSC advances, 2016, 6(80): 76551-76555.
- [26] Late D J. Temperature dependent phonon shifts in few-layer black phosphorus[J]. ACS applied materials & interfaces, 2015, 7(10): 5857-5862.
- [27] Erande M B, Pawar M S, Late D J. Humidity sensing and photodetection behavior of electrochemically exfoliated atomically thin-layered black phosphorus nanosheets[J]. ACS applied materials & interfaces, 2016, 8(18): 11548-11556.
- [28] Bhorde A, Pawbake A, Sharma P, et al. Solvothermal synthesis of tin sulfide (SnS) nanorods and investigation of its field emission properties[J]. Applied Physics A, 2018, 124(2): 133.
- [29] Late D J. Liquid exfoliation of black phosphorus nanosheets and its application as humidity sensor[J]. Microporous and Mesoporous Materials, 2016, 225: 494-503.
- [30] Suryawanshi S R, More M A, Late D J. Laser exfoliation of 2D black phosphorus nanosheets and their application as a field emitter[J]. RSC advances, 2016, 6(113): 112103-112108.

Detailed Responses to the Reviewer Comments-2

1) The discussion part is not strong enough, most of the articles are explaining the data. It's really difficult to persuade the author's point of view, especially for the lubricating mechanism.

Response: Thank you for your suggestion. We have revised the discussion part and enriched the lubricating mechanism. According to Hamrock-Dowson (H-D) theory, the calculated ratio of film thickness to surface roughness λ is approximately 0.16, is smaller than 1, and thereby indicating the current lubrication is in the regime of boundary lubrication. The revised parts are marked the red front.

The added and revised parts are presented as follows.

According to classical lubrication theories, the lubrication regimes can be identified in a lubrication regime map in terms of the two variables (g_v and g_E):

$$g_v = \frac{GW^3}{U^2} \quad (6)$$

$$g_E = \frac{W^{8/3}}{U^2} \quad (7)$$

where $U = \eta V/E'R$, $G = \alpha E'$, $W = F/E'R^2$, V is the averaged linear velocity (62.8mm/s), η is the bulk viscosity of lubricating solution, α is the viscosity-pressure coefficient, R is the radius of the ball (3mm), E' is the effective modulus of the elasticity (162GPa), F is the normal load (15N), k (≈ 1.03) is the elliptical parameter. According to Hamrock-Dowson (H-D) theory, the minimum film thickness can be calculated using following formula [37]:

$$h_{\min} = 2.69 \frac{G^{0.53} U^{0.67}}{W^{0.067}} (1 - 0.61 e^{-0.73k}) \quad (8)$$

The lubrication regime can be determined by the ratio of theoretical minimum film thickness to the combined surface roughness. The following formula can be used to calculate the ratio γ :

$$\gamma = \frac{h_{\min}}{\sqrt{\sigma_1^2 + \sigma_2^2}} \quad (9)$$

where σ_1 and σ_2 are the roughness of the opposite rubbing surfaces after the lubrication tests. ($\sigma_1 = 40\text{nm}$, $\sigma_2 = 20\text{nm}$)

Finally, the calculated h_{\min} is about 7.54 nm when the load is 15 N and the speed is 62.8mm/s. Thus, the calculated ratio of film thickness to surface roughness λ is approximately 0.16, is smaller than 1, and thereby indicating the current lubrication is in the regime of boundary lubrication [38]. Therefore, it can improve the tribological performance of LP it can be concluded that the electrostatic adsorption and interlayer shear on the interface. BP nanosheets play an important role in the friction-reducing ability of BP-LP lubricants.

2) It is recommended to calculate wear from the profile of the wear scar, so that the mechanism of wear can be more accurately observed.

Response: Thank you for your suggestion. We have given the Hertz contact model of sphere-on-disc in figure 7, according to calculate corresponding maximum Hertzian contact stress, made some detailed explanations about the mechanism of wear. The revised parts are marked the red front.

The added and revised parts are presented as follows.

The above phenomenon can be explained by the Hertz's theory [31]. The tribological properties of BP-LP lubricants were investigated by sphere-on-disc friction tests. Sphere contact as object, contact stress formula of Hertz are deduced theoretically from the contact model of spheres and disc [32], as shown in Fig. 7. When the sphere is in contact with disk under the external loading, elliptic contact area will be produced around the contact point due to the partial deformation of sphere and disc.

The corresponding contact stresses were calculated according to Eq. (1).

$$q_0 = \frac{4F}{\pi a^2} \quad (1)$$

$$a = 2 \left(\frac{2}{3} * \frac{FR}{E'} \right) \quad (2)$$

Where, q_0 and a are maximum Hertz contact stress and Hertz contact diameter, respectively. F is the normal load (8N, 10N, 12N, 15N), R is the radius of the ball (3mm), E' is the effective elastic modulus. It can be expressed by the following formula [33]:

$$\frac{1}{E'} = \frac{1}{2} \left(\frac{1 - \mu_1^2}{E_1} + \frac{1 - \mu_2^2}{E_2} \right) \quad (3)$$

where E_1 (110 GPa) and E_2 (210 GPa) are the elastic moduli of the TC4 and GCr15,

respectively. And μ_1 (0.34) and μ_2 (0.30) are the poisson ratios of the TC4 and GCr15, respectively. Thus, the corresponding maximum Hertzian contact stress are 1039MPa, 1119MPa, 1190MPa and 1281MPa. As for the Eq. (2), the elliptic contact area is increased proportional to the F , which indicates that the contact area increased more slowly than that of load. Based on the formulae of the friction coefficient, $\mu = f/F$ (μ is the COF, f is the frictional force, F is the load), it can be concluded that the values of the COF may be reduced as the load increased. The wear volumes of the ball side are calculated by the following formula [34]:

$$V = \left(\frac{\pi l}{6}\right)\left(\frac{3d^2}{4} + l^2\right) \quad (4)$$

$$l = r - \sqrt{r^2 - \frac{d^2}{4}} \quad (5)$$

V is the wear volume, r is the radius of the ball, d is the wear scar diameter.

Hence, the wear rate was decreased as the load increased and reduced the wear. These results showed that adding BP nanosheets as a lubrication additive into liquid paraffin can significantly improve the lubricating properties of the base oil of liquid paraffin and effectively reduce the wear of the GCr15/TC4 friction pair.

Fig. 7 Hertz contact model of sphere-on-disc